# Mitigating Dietary Microplastic Accumulation and Oxidative Stress Response in European Seabass (*Dicentrarchus labrax*) Juveniles Using a Natural Microencapsulated Antioxidant

**DOI:** 10.3390/antiox13070812

**Published:** 2024-07-05

**Authors:** Matteo Zarantoniello, Nico Cattaneo, Federico Conti, Margherita Carrino, Gloriana Cardinaletti, İdris Şener, Ike Olivotto

**Affiliations:** 1Department of Life and Environmental Sciences, Università Politecnica delle Marche, 60131 Ancona, Italy; n.cattaneo@pm.univpm.it (N.C.); f.conti@pm.univpm.it (F.C.); marghecarrino99@gmail.com (M.C.); idris_943@hotmail.com (İ.Ş.); 2Department of Agricultural, Food, Environmental and Animal Sciences, University of Udine, 33100 Udine, Italy; gloriana.cardinaletti@uniud.it

**Keywords:** natural astaxanthin, natural-based microcapsules, starch, microplastic coagulation, fish welfare, feeding trial, European seabass

## Abstract

Aquafeed’s contamination by microplastics can pose a risk to fish health and quality since they can be absorbed by the gastrointestinal tract and translocate to different tissues. The liver acts as a retaining organ with the consequent triggering of oxidative stress response. The present study aimed to combine the use of natural astaxanthin with natural-based microcapsules to counteract these negative side effects. European seabass juveniles were fed diets containing commercially available fluorescent microplastic microbeads (1–5 μm; 50 mg/kg feed) alone or combined with microencapsulated astaxanthin (AX) (7 g/kg feed; tested for half or whole feeding trial—30 or 60 days, respectively). Fish from the different dietary treatments did not evidence variations in survival and growth performance and did not show pathological alterations at the intestinal level. However, the microplastics were absorbed at the intestinal level with a consequent translocation to the liver, leading, when provided solely, to *sod1*, *sod2*, and *cat* upregulation. Interestingly, the dietary implementation of microencapsulated AX led to a mitigation of oxidative stress. In addition, the microcapsules, due to their composition, promoted microplastic coagulation in the fish gut, limiting their absorption and accumulation in all the tissues analyzed. These results were supported by in vitro tests, which demonstrated that the microcapsules promoted microplastic coagula formation too large to be absorbed at the intestinal level and by the fact that the coagulated microplastics were released through the fish feces.

## 1. Introduction

Plastic particles under 5 mm in size, defined as microplastics, are widely recognized as emerging contaminants [1] that are globally distributed from terrestrial to aquatic ecosystems [2,3,4] at all trophic levels [5]. Aquaculture, a rapidly growing industry that produces food from aquatic sources, is susceptible to contamination from microplastics [6,7]. The primary sources of MPs in fish farming, whether in freshwater or marine environments, include environmental pollution, the use of plastic tools, and fish feeds [8,9,10]. In reference to aquafeeds, the raw materials, processing, and packaging procedures all contribute to the unintentional introduction of microplastics into the diets of fish [11,12].

Although it has been demonstrated that fish are able to distinguish microplastics as inedible particles [13], MPs become more challenging to detect when they are incorporated into the feed, resulting in increased ingestion by fish [14,15,16]. The impact of microplastic toxicity can vary depending on their size, shape, chemical composition, concentration, and exposure time [17,18,19,20,21,22], as well as the fish species or their life-cycle stage [23,24,25,26]. When ingested by fish, the major adverse effects include gastrointestinal tract obstruction, alterations in feeding behavior, reduced growth rates, decreasing swimming activity, inflammations, alteration in gut microbiota, and induction of immune, stress, and oxidative stress responses [27,28,29,30,31,32]. Fish have biological barriers that help to mitigate these side effects. In fact, at the intestinal level, they can only uptake microplastic particles that are less than 20 µm in size, with larger particles typically being expelled through their feces [33,34,35]. Differently, smaller microplastics typically pass through the intestinal epithelium, enter the bloodstream [36,37], and often accumulate in the liver [38]. Eventually, trace amounts of microplastics can reach the muscular tissue of fish [39,40], as evidenced by their presence in fillets of commercially farmed species such as European seabass (*Dicentrarchus labrax*) [9,10,41], gilthead seabream (*Sparus aurata*) [42,43], and Atlantic salmon (*Salmo salar*) [44]. Microplastics accumulated in the liver parenchyma have been associated in many fish species with an increased oxidative stress response and an alteration of animal welfare, while their detection in fish muscle poses concern for human health [45,46,47,48].

Although the ingestion of microplastics and their associated harmful effects on fish have been extensively investigated, only a few studies have focused on strategies to mitigate their effects on fish. In the experimental model zebrafish (*Danio rerio*), the dietary implementation of vitamin D has been demonstrated to be able to alleviate the immunotoxicity, neurotoxicity, and lipid metabolism disorders induced by microplastics (80 nm; 15 or 150 µg/L) ingestion [49,50]. As regards finfish species, the dietary inclusion of probiotics solely was able to reverse the liver disorders and restore the antioxidant capacity in Nile tilapia (*Oreochromis niloticus*) exposed to 0.5 µm microplastics (1 mg/L) [51], while the combination of probiotics and vitamin C was able to reduce gill, hepatic, and intestinal damages caused by microplastic exposure in the same species [52,53,54].

In addition, the dietary inclusion of lycopene, citric acid, or the microalga *Chlorella vulgaris* counteracted the negative effects of dietary microplastics (size > 100 nm at 500 mg/kg of feed) in African catfish (*Clarias gariepinus*), ameliorating hematological parameters, enhancing the antioxidant activity, and reversing most of the histological alterations detected in brain, liver, kidney, and intestine [55,56,57]. Lycopene and *C. vulgaris* dietary administration have also been reported to be able to restore microplastic-induced reproductive dysfunctions in the same species [58]. As eliminating microplastics from the environment and aquafeeds is nearly impossible, it is essential to implement efficient mitigation techniques to tackle these challenges and guarantee the safety and sustainability of seafood production. To cope with oxidative stress, the use of natural antioxidant molecules like fruit by-products, medicinal herbs, ferulic acid, bioactive peptides, ascorbic acid, α-tocopherol, sulfated polysaccharides, and carotenoids can represent a proper strategy [59,60,61,62,63,64]. Among carotenoids, astaxanthin (AX) has been extensively used in aquaculture [65,66] not only for its antioxidant properties but also for additional beneficial effects on farmed species, such as enhanced pigmentation, improved survival and growth rates, increased reproductive success, and greater disease resistance [67,68,69,70]. Specifically, natural AX, mainly extracted from *Haematococcus pluvialis* [71], should be chosen with respect to its synthetic counterpart because of its increased antioxidant efficacy [72,73,74] despite its higher production cost [75,76]. However, it has to be pointed out that natural AX shows rapid degradation due to its low stability, particularly when exposed to industrial conditions [77,78], possibly limiting its application as an aquafeed supplement [79].

To protect AX from degradation, a number of microencapsulation techniques have recently emerged, including lyophilization, co-crystallization, hydrogel formation, high-pressure homogenization, lipid nano-dispersion, liposomes incorporation, and spray-drying [80,81]. The microencapsulation technology thus represents an excellent solution for maintaining AX integrity and stability over time, enhancing its resistance to high temperatures, light, and oxygen exposure [82,83,84]. Natural AX can thus be converted into a dry powder enclosed in microcapsules composed of a number of natural substances, such as starch, Arabic gum, chitosan, or maltodextrin [80,81,85,86,87,88]. 

Among these natural substances used to produce microcapsules, some have recently shown the ability to coagulate water-suspended microplastics [89,90,91,92]. The coagulation process, previously mediated by metal ions and today by natural molecules, involves the formation of larger flocs of microplastics whose size facilitates their removal through filtration or sedimentation [93,94,95,96]. Particularly, starch is known for its chemical properties, affordability, availability, and biodegradability and has also garnered interest as an organic and sustainable substance for coagulating water-suspended microplastics [97,98]. Within this context, the primary objective of the present study was to combine the use of natural AX protected in microcapsules characterized by a natural-based wall matrix, including starch, in the rearing of European seabass juveniles fed diets containing fluorescent microplastic microbeads. Microcapsules have been developed to perfectly adhere to aquafeeds and facilitate their delivery to the fish. Results showed that this innovative approach has the capability to decrease the absorption of microplastics by fish due to the coagulation processes happening in the gut while reducing fish oxidative stress because of the natural antioxidant release. A comprehensive understanding of microplastic assimilation, fate, and fish welfare in response to dietary microencapsulated natural AX was achieved through a multidisciplinary laboratory approach.

## 2. Materials and Methods

### 2.1. Ethical Standards

All the procedures involving animals conducted in the present study were conducted according to EU legal frameworks relating to the protection of animals used for scientific purposes (Directive 2010/63/EU). This study was approved by the Ethics Committee of the Marche Polytechnic University (Ancona, Italy; n.3 24/11/2022) and the Italian Ministry of Health (Aut. n. 391/2023-PR). The animals’ discomfort was minimized using an anesthetic (MS222; Merck KGaA, Darmstadt, Germany).

### 2.2. Microplastics’ Characteristics

Amino formaldehyde polymer (company code: FMV-1.3; MPs) fluorescent microbeads (1.22 × 10^11^ microbeads/g), characterized by a size range of 1–5 µm (d_50_ = 1.5–2.0 µm, d_95_ = 3.5 µm, d_99_ = 5 µm) and an emission peak of 636 nm when excited at 584 nm, were purchased from Cospheric LLC (Goleta, CA, USA; https://www.cospheric.com/fluorescent_violet_tracer_microparticles_2um.htm; accessed on 25 February 2022).

### 2.3. Astaxanthin and Microcapsules’ Composition

The natural AX (AstaReal^®^ L10, Nacka, Sweden) was encapsulated by STM Aquatrade S.r.l. (Castel Raimondo, Macerata, Italy) utilizing a novel technology named Co.M.E. (Coating Made Easy), which was created as part of a private project called +POP (Powder on Pellets; STM Aquatrade S.r.l). The distinctive microcapsule wall matrix consisted primarily of Arabic gum (55%), starch (22%), and, to a lesser extent, cellulose, sodium ascorbate, and vitamin E, which were ranked in descending order of prevalence. The average size of microcapsules AX was 47.51 ± 7.23 µm (Figure 1a), and the surface appeared smooth and uniform (Figure 1b). Due to intellectual property protection, the specifics of the microcapsule preparation process cannot be fully disclosed. This encapsulation method not only safeguards the included AX molecule but also maximizes its release into the feed. The technology facilitates strong adherence of the dry microcapsules to aquatic feed particles through precise physical and chemical interactions. Due to their chemical features, the microcapsules release their contents within approximately 90 s upon contact with water. The provision of 1 g of AX microcapsules corresponds to a net transfer of 25 ppm of AX to the feed (equivalent to 2.5% of the filled microcapsule composition).

### 2.4. Experimental Diets

All the diets were formulated and prepared at the Department of Agriculture, Food, and Environmental and Animal Science of University of Udine (Italy), starting from identical batches of individual ingredients. A control diet (named C, free of fluorescent MP microbeads) was formulated to match the proximate composition of the successful control diet recently used in a previous study on European seabass juveniles [99]. The MP diet was prepared by including the fluorescent MP microbeads in the control mixture diet at a concentration of 50 mg/kg feed. Before being included in the preparation of the MP-containing diets, fluorescent microbeads, hydrophobic in their pristine state, were resuspended in a tween-80 (Merck KGaA) solution (0.1%) as a surfactant and subsequently washed three times using deionized water, according to the company’s technical support suggestion.

All powdered ingredients used to produce the test diets were well mixed (GastroNorm 30C1PN, ItaliaGroup Corporate Srl, Ponte nelle Alpi, Italy) for 20 min, adding oil and water into the blend to achieve the appropriate consistency suitable for pelleting. Water was then used to include fluorescent MP microbeads in the mixture. Using a 3 mm die meat grinder, pellets were obtained and dried at 37 °C for 48 h in a ventilated heater and then vacuum-stored.

To confirm the absence of eventual contamination of fluorescent MP microbeads in the control diet (C), three subsamples were checked through a confocal microscope (Nikon A1R; Nikon Corporation, Tokyo, Japan). In addition, three subsamples were subjected to chemical digestion followed by filtration on fiberglass filters with 0.7 µm-pores (Whatman GF/A, Merck KGaA). The filters were dried at room temperature and analyzed through a Zeiss Axio Imager.A2 (Zeiss, Oberkochen, Germany) to confirm the absence of fluorescence MP microbeads (for details on the procedure, please refer to the dedicated section).

Starting from the above-described diets (C and MP), the following ones were prepared as follows: (i) C+AX diet: 7 g/kg feed of AX microcapsules (corresponding to 175 mg/kg of AX) were added to the control diet (C); (ii) MP+AX diet: 50 mg/kg feed of fluorescent MP microbeads and 7 g/kg feed of AX microcapsules (corresponding to 175 mg/kg of AX) were added to the control diet (C).

To produce these diets, 1 kg batch of C or MP diets and 7 g of microencapsulated AX were transferred to glass airtight jars and then vigorously mixed to allow them to uniformly cover the feed pellets.

The effective presence of MPs and/or AX microcapsules in the experimental diets was checked by analyzing feed subsamples (in triplicate for each diet) by a confocal microscope (Nikon A1R; Nikon Corporation) or a stereomicroscope (Leica, Wetzlar, Germany) equipped with a camera (Invenio 10SCIII, DeltaPix; Smoerum, Denmark), respectively. The aim of this study was to exclusively assess the effects of voluntarily added fluorescent MP microbeads during MP diet preparation, excluding the potential intrinsic microplastic contamination of the ingredients used. It was assumed that this level of contamination was consistent among all the tested diets since the basic mixture was composed of the same ingredients from the same batches. Therefore, any effect observed in fish was uniquely attributable to the added fluorescent MP microbeads.

Finally, the dietary AX content determination was performed on 3 subsamples of C+AX and MP+AX diets analyzed at the beginning and at the conclusion of the feeding trial, according to Du et al. [100]. An average concentration of 172 ± 6 mg/kg for both diets was detected at both sampling times.

### 2.5. In Vitro Microplastic Coagulation

To assess their potential role in MP coagulation, whole microcapsules (empty or filled with AX), Arabic gum, and starch (as main microcapsules’ components) were tested in an in vitro experiment.

A stock solution was prepared by adding fluorescent MP microbeads at a concentration of 50 mg/L to salt water (salinity: 30‰), and pH was adjusted to 3 by using HCl 0.1 M to mimic the conditions of European seabass stomach after the feed ingestion [101].

This MP stock solution was divided into different aliquots in which AX microcapsules, AX alone, empty microcapsules, Arabic gum, or starch were added at the same concentration present in the feeds used in the current study, considering the relative composition of the microcapsules: (i) 7 g/L for AX microcapsules; (ii) 0.175 g/L for AX alone since it represents the 2.5% of the filled microcapsule composition; (iii) 6.825 g/L for empty microcapsules; (iv) 3.850 g/L for Arabic gum since it represents the 55% of the microcapsule wall matrix; (v) 1.540 g/L for starch since it represents the 22% of the microcapsule wall matrix. An additional aliquot was maintained with only fluorescent MP microbeads (50 mg/L).

From each of the above-mentioned aliquots, 1 mL (in triplicate; n = 3) was placed in a 35 mm glass bottom dish (GmbH, Gräfelfing, Germany) and observed under a Nikon A1R confocal microscope (Nikon Corporation, Tokyo, Japan) prior and after a 6 h period on a shaker plate (SO3 Orbital Shaker; Stuart Scientific Co., Ltd., Redhill, England) to mimic fish stomach movements and the gastric evacuation time of European seabass [102]. At both observation steps, all the solutions were excited with 561 nm wavelength (collecting emissions at 615 nm) to visualize the fluorescent MP microbeads in red through a Nikon A1R confocal microscope (Nikon Corporation). The eventual occurrence of coagulation events was verified, and coagula, if present, were measured. The images were processed with NIS-Element software (version 5.21.00, Nikon).

After this first step, all the 35 mm glass bottom dishes were subjected to a pH transition from 3 to 6 by adding NaOH 1M to reach the average pH detected in European seabass anterior intestine [101]. After an additional 4 h period on a shaker plate (SO3 Orbital Shaker), all the solutions were again excited with 561 nm wavelength (collecting emissions at 615 nm) using a Nikon A1R confocal microscope (Nikon Corporation) to verify the eventual occurrence of coagulation events and to measure them (image processing NIS-Element software).

### 2.6. Experimental Design

Five hundred and forty European seabass juveniles (initial body weight 74.3 ± 2.0 g) provided by Panittica Pugliese (Torre Canne di Fasano, Brindisi, Italy) were subjected to a 2-week acclimation period in a 2000 L tank with all the filtration equipment (Panaque s.r.l., Viterbo, Italy) to temperature of 17.0 ± 0.5 °C, salinity of 30 ± 0.5‰, ammonia and nitrite concentration under 0.05 mg/L, and nitrate concentration under 10 mg/L. Then, fish were randomly divided into six experimental groups as follows: (i) C group: fish fed the control diet (C) for the whole trial (60 days); (ii) C/C+AX group: fish fed the control diet (C) for the first 30 days and the C+AX diet for the following 30 days; (iii) C+AX group: fish fed the C+AX diet for the whole trial (60 days); (iv) MP group: fish fed the MP diet for the whole trial (60 days); (v) MP/MP+AX group: fish fed the MP diet for the first 30 days and the MP+AX diet for the following 30 days; (vi) MP+AX group: fish fed the MP+AX diet for the whole trial (60 days).

Each experimental group (60 fish per group; 20 fish per tank) consisted of three 500 L tanks provided with their own mechanical and biological filtration (Panaque). During the whole trial, fish were subjected to a natural photoperiod (12 light/12 dark), and a constant water temperature in the tanks was maintained (17.0 ± 0.5 °C); a visual inspection of each tank was carried out daily to check the eventual presence of dead specimens. Almost 30% of the water was changed every day, and mechanical filters were cleaned daily. Fish were hand-fed the experimental diets (pellet size: 3 mm) at 1.5% body weight (half ration in the morning and half ration in the afternoon). Every two weeks, the daily feed amount was adjusted by weighing all the fish in each tank. All the provided feed was completely consumed by the fish in about 15 min after administration.

For the last 10 days of the feeding trial, 3 fish per tank were transferred daily (in the morning, after being fed) to a 100 L plastic tank, which was floating in the main tank, equipped with lateral 0.5 cm holes (in the upper part of the tank), covered by a net to ensure water exchange, and had a double floor (consisting of a 5 mm grid suspended at 5 cm from the plastic floor) to allow feces collection. After 8 h, fish were returned to their main tank, and feces were collected. Three samples of feces were collected daily for each experimental group and pooled together, resulting in 10 samples per experimental group at the end of the trial. Feces samples were incubated in open 1.5 mL Eppendorf tubes at 40 °C overnight in an Argo Lab ICN 35 incubator (Argo Lab, Carpi, Italy) for water evaporation and then stored at −20 °C until the chemical digestion (for details, refer to the dedicated section).

At the end of the feeding trial, all the fish were individually weighed after being euthanized (MS222, 1 g/L; Merck KGaA), and then. Data were used to calculate (i) weight gain: WG (g) = final body weight − initial body weight; (ii) relative growth rate: RGR (%) = [(final body weight − initial body weight)/initial body weight] × 100; and (iii) specific growth rate: SGR (%) = [(ln final body weight − ln initial body weight)/time (days)] × 100. Survival rate was calculated as follows: SR (%) = (final number of fish/initial number of fish) × 100.

Samples of blood (collected from the caudal vein using a heparinized syringe), muscle, adipose tissue, distal intestine, and liver were collected and adequately preserved for further analyses, as reported in the dedicated sections.

### 2.7. Confocal Microscopy for Microplastic Detection in Fish Tissues

Nikon A1R confocal microscope (Nikon Corporation, Tokyo, Japan) was used to demonstrate the presence of fluorescent MP microbeads in blood, muscle, adipose tissue, liver, and distal intestine samples collected from 5 fish per tank (15 fish per experimental group; n = 15). Two wavelengths of 561 and 647 nm were simultaneously used to excite samples, and the related emissions were collected at 615 and 670 nm to visualize the fluorescent MP microbeads in red and (chosen pseudo-color for the far-red) the tissue texture in blue. Finally, NIS-Element software (version 5.21.00; Nikon) was used to elaborate the acquired images.

### 2.8. Microplastics’ Quantification in Feed and Fish Tissues and Feces

Three subsamples per diet sample (n = 3) of blood, muscle, adipose tissue, liver, and distal intestine from 5 fish per tank (15 fish per experimental group; n = 15), as well as samples of feces collected during the last 10 days of the trial (n = 10), were weighed and subjected to a 10% KOH-mediated digestion process, according to Cattaneo et al. [20]. Briefly, each sample was located in a glass tube in which the 10% KOH solution (1:10 *w*/*v* ratio) was added and incubated for 48 h at 40 °C. After this period, a vacuum pump connected to a filter funnel was used to filter each sample’s digestate, using 0.7 µm pore size fiberglass filters (Whatman GF/A, Merck KGaA) that were then dried at room temperature and stocked in glass Petri dishes. The MP quantification analyses were conducted using a Zeiss Axio Imager.A2 (Zeiss, Oberkochen, Germany) with Texas Red (561 nm) and FITC (491 nm) channels. Finally, the ZEN Blue 2.3 software (Zeiss) was used to manually count the fluorescent MP microbeads on each filter.

### 2.9. Distal Intestine and Liver Histology

Samples of liver and distal intestine were collected from 5 fish per tank (15 fish per experimental group; n = 15) and fixed for 24 h at 4 °C by immersion in Bouin’s solution (Merck KGaA). Samples were then processed as described in Zarantoniello et al. [99] to obtain section of 5 μm in thickness. Particularly, for each sample, three transversal sections collected at intervals of 200 μm were considered for the morphometric and histopathological evaluations.

Specifically, for distal intestine, sections were stained with Mayer’s hematoxylin and eosin Y (HE; Merck KGaA, Darmstadt, Germany) to determine height and eventual presence of fusion episodes of mucosal folds, presence of supranuclear vacuoles in the enterocytes, submucosa width, and lymphocyte infiltration at submucosal level as indicator of inflammatory processes. Alcian blue (Bio-Optica, Milano, Italy) staining was instead used to calculate the relative abundance of Alcian blue positive (Ab+) goblet cells on a 500 µm^2^ surface area. Table 1 reports the score assignment criteria for the above-mentioned parameters, according to Zarantoniello et al. [99]. Differently, for liver, HE staining was used to assess hepatocyte morphology and the presence of pathological alterations. Images were acquired using a combined color digital camera (Axiocam 105, Zeiss), and measurements were taken with the ZEN 2.3 software (Zeiss).

### 2.10. Gene Expression Analysis

The relative mRNA abundances of target genes in liver and distal intestine samples from 5 fish per tank (15 per experimental group) were measured using real-time quantitative polymerase chain reaction (qPCR). Total RNA extraction, cDNA synthesis, and qPCRs were conducted as previously described in Zarantoniello et al. [99]. The melting curve revealed one single peak in each run, confirming the specificity of the PCR products, and no peaks were detected for the two no-template controls added in each run. Relative quantification of the expression of genes involved in immune response (interleukin-1β, *il1b*; interleukin-10, *il10*; tumor necrosis factor alpha, *tnfa*) was performed on distal intestine samples. Relative quantification of the expression of genes involved in oxidative stress response (superoxide dismutase 1, *sod1*; superoxide dismutase 2, *sod2*; catalase, *cat*) was performed on liver samples. The primer sequences are summarized in Table 2. The mRNA levels of target genes analyzed were calculated through the software iQ5 optical system version 2.0 (Bio-Rad, Hercules, CA, USA) and the GeneEx Macro iQ5 Conversion and GeneEx Macro iQ5 files, using the expression of two reference genes (beta-actin, *β-actin*; 18S ribosomal protein, *18s*) in both tissues.

### 2.11. Statistical Analyses

The tanks were used as the experimental unit for data related to survival rate and zootechnical performance, while fish were considered the experimental unit for all the remaining analyses. Normality and homoscedasticity were checked for all the data through Shapiro–Wilk and Levene’s tests, respectively. Data were then analyzed using a one-way ANOVA followed by Tukey’s multiple comparison test. Significance was set at *p* < 0.05, and results were expressed as mean ± standard deviation (SD). The statistical software package Prism-8 (GraphPad Software, San Diego, CA, USA) was used for the data analysis.

## 3. Results

### 3.1. In Vitro Microplastics’ Coagulation

At pH 3, no coagulation events were identified in the solution containing only fluorescent MP microbeads, as well as in the solutions containing microplastics combined with Arabic gum or AX (Figure 2a–c). Differently, in all the microplastic solutions containing AX microcapsules, empty microcapsules, or starch, coagulation events were noted (Figure 2d–f). The average sizes of the coagula were 65.4 ± 7.7, 58.9 ± 9.3, and 52.5 ± 8.7 µm in the solutions containing AX microcapsules, empty microcapsules, or starch, respectively.

When the pH was changed to 6, all the coagula detected in the microplastic solutions containing AX microcapsules, empty microcapsules, or starch remained intact (Figure 2g–i). Additionally, no coagulation events were observed in the solutions containing microplastics alone or in combination with Arabic gum or AX.

### 3.2. Fish Survival Rate and Growth Indexes

During the feeding trial, no dead specimens were found in all the experimental groups (100% survival rate; SR). Growth indexes are reported in Table 3. Final body weight (FBW), relative growth rate (RGR), and specific growth rate (SGR) did not show significant differences among the experimental groups.

### 3.3. Microplastic Detection in Fish Tissues

No fluorescent MP microbeads were observed in any of the tissues analyzed in fish from C, C/C+AX, and C+AX groups. Differently, fluorescent microbeads were detected in all the tissues sampled from fish fed diets containing MPs. Particularly, in both MP and MP/MP+AX groups, fluorescent microbeads were found in the intestinal mucosa (Figure 3a–c), in the hepatic parenchyma (Figure 4a), dispersed among the blood cells (Figure 4b), and, to a lesser extent, in both muscle (Figure 4c) and adipose tissue (Figure 5a,b). Differently, considering the MP+AX group, fluorescent MP microbeads were found in all the analyzed tissues except for muscle.

### 3.4. Microplastics’ Quantification in Feed and Fish Tissues and Feces

The quantification analyses on feed samples confirmed the absence of fluorescent MP microbead contamination in both C and C+AX diets as well as their presence in comparable amounts in both MP and MP+AX diets (6148 ± 124 and 6134 ± 107 fluorescent MP microbeads/mg feed, respectively; *p* > 0.05).

Considering fish tissues (Table 4), the quantification of fluorescent MP microbeads (per gram of tissue) in samples taken from the intestine, liver, muscle, adipose tissue, and blood is reported. None of the tissues sampled from fish in the C, C/C+AX, and C+AX groups contained fluorescent MP microbeads.

As regards fed diets containing MPs, both MP/MP+AX and MP+AX groups showed significantly (*p* < 0.05) lower levels of fluorescent microbeads in the intestine and blood samples compared to the MP group. Notably, the MP+AX group had the lowest quantification values in these tissues (although it was not statistically significant compared to the MP/MP+AX group in the intestine). In the liver, muscle, and adipose tissue samples of the MP+AX group, there were significantly (*p* < 0.05) lower levels of quantification (with no MPs found in muscle tissue) compared to the MP and MP/MP+AX groups, which did not show significant differences between them.

Finally, considering the quantification analyses on feces samples (Table 4), a significantly (*p* < 0.05) lower abundance of fluorescent MP microbeads was detected in the MP group compared to MP/MP+AX and MP+AX groups, which did not show significant differences between them.

### 3.5. Histological Analysis

Fish from all the experimental groups did not evidence pathological alterations in both the distal intestine and liver. The proper architecture of these organs was unaltered by the different dietary treatments. As regards the distal intestine, no significant differences were detected among the experimental groups in terms of mucosal fold height, submucosa width, episodes of mucosal fold fusion, presence of basal inflammatory influx, and Ab+ goblet cells’ relative abundance (Table 5). However, all the fish fed MP-containing diets were characterized by the presence of a dense supranuclear vacuolization in the apical part of enterocytes (Figure 6e–g), which was scattered in fish from C, C/C+AX, and C+AX groups (Figure 6a–d). In particular, while the MP and MP/MP+AX groups showed a highly abundant supranuclear vacuolization, fish from the MP+AX group were characterized by a reduction of these vacuoles (refer to both Figure 6e and Table 5).

Considering the liver, all the experimental groups showed a physiological structure of hepatic parenchyma, and no appreciable inflammation signs were detected. Fish from all the experimental groups were characterized by a modestly fat liver parenchyma with a diffuse presence of hepatocytes with cytoplasm filled with fat (Figure 7).

### 3.6. Real-Time qPCRs

Considering the expression of genes involved in the immune response (*il1*, *il10*, and *tnfa*; Figure 8a–c) analyzed in distal intestine samples, no significant differences were evident among the experimental groups.

As regards the expression of genes involved in oxidative stress response (*sod1*, *sod2*, and *cat*; Figure 8d–f) analyzed in liver samples, no significant variations were detected among C, C/C+AX, and C-AX groups. Differently, both the MP/MP+AX and MP+AX groups were characterized by a significantly (*p* < 0.05) lower *sod1*, *sod2*, and *cat* relative expression compared to the MP group, showing values comparable to those observed in all the control groups (except for the MP/MP+AX group, which had a higher *cat* expression).

## 4. Discussion

The European seabass exposed to the different dietary treatments of the present study did not evidence variations in survival and growth performance, with the results being consistent with a previous study and demonstrating that the administration of dietary microplastic did not reduce growth in gilthead seabream after a 45-day exposure time [103]. Furthermore, all the groups that were fed MP diets did not exhibit any pathological alterations or signs of inflammation in the distal intestine. This was evidenced by both the histological indexes and the relative expression of immune response markers (*il1b*, *il10*, and *tnfa*) that were analyzed, suggesting that the transit of small microplastics like those used in the present study did not alter the intestinal structure or its functionality, as already demonstrated in different fish species [20,104]. Accordingly, it has been demonstrated that big particles (size range between 200 and 1000 µm), however, are well known to cause severe intestine alterations [105].

Microplastics lower than 20 µm in size can be absorbed at the intestinal level and can thus be translocated to other organs, mainly in the liver. In this organ, the microplastic accumulation causes oxidative stress, which is one of the most detrimental effects associated with fish exposure to this contaminant [38]. Oxidative stress can result in an impairment of cellular processes, including protein denaturation, lipid peroxidation, and apoptosis [106,107], which may consequently affect the overall health and quality of farmed fish over the long term [108].

Due to the microplastics’ wide distribution, oxidative stress must be considered a diffuse issue for the aquaculture sector, with potential impacts on fish welfare and economic value. In this context, the implementation of aquafeeds with antioxidants like AX can be considered a promising solution since the use of this molecule to mitigate oxidative stress resulting from microplastic exposure has recently gained attention. Studies have shown that natural AX can help counteract inflammatory responses triggered by microplastic exposure in the head kidney cells of Nile tilapia (*Oreochromis niloticus*) [109] and European carp (*Cyprinus carpio*) [110]. Additionally, when AX was added to the water at a concentration of 30 nM, it partially reversed oxidative stress and reduced cardiovascular toxicity in zebrafish early larvae. This was achieved by scavenging reactive oxygen species and inhibiting inflammation when fish were exposed to a combination of water contaminated with micro- and nano-plastics along with microcystin-LR from 3 to 168 h post fertilization [111]. However, these studies did not provide a comprehensive overview of the possible application of AX to counteract the negative side effects of dietary microplastic administration in fish.

In the present study, the dietary administration of fluorescent MP microbeads (size of 1–5 µm, concentration of 50 mg/kg) led to intestinal absorption with a consequent translocation to other tissues/organs, mainly the liver. Previous research has suggested that the fish liver serves as a significant organ for retaining microplastics, leading to a notable accumulation within the tissue [112,113]. In fact, in all the groups fed MP-implemented diets, the liver represented the major site of accumulation, as suggested by the quantification analyses, confirming its role in trapping the fluorescent MP microbeads used and preventing/reducing their translocation to other target organs/tissues. However, this accumulation may trigger the activation of oxidative stress defense mechanisms [112,113]. Accordingly, in the present study, fish from the MP group exhibited a significantly higher relative expression of *sod1*, *sod2*, and *cat* compared to other experimental groups despite no structural alterations being found in the liver tissue architecture. Interestingly, fish from the MP+AX group that were fed diets implemented with AX microcapsules throughout the entire 60-day trial showed relative mRNA abundance levels similar to those detected in control groups for all these markers. Furthermore, fish belonging to the MP/MP+AX group, which were exposed to microencapsulated AX for a mere 30-day period, only displayed mitigation in the expression of oxidative stress markers. Taken together, these results suggested that the intervention may have a dose- and/or time-dependent effect. This final finding was also supported by the observations of Huang et al. [114], which evidenced that the dietary inclusion of AX (200 mg/kg) cannot completely counteract the oxidative stress caused by environmental microplastics in discus fish (*Symphysodon aequifasciatus*). In this regard, additional research regarding the best administration time and concentration of this antioxidant is needed. However, the *sod1*, *sod2*, and *cat* relative expression observed in the present study suggested the effectiveness of natural AX, preserved through the microencapsulation technology, in mitigating the oxidative stress response associated with aquafeed microplastic contamination.

Another interesting result highlighted by the present study is related to the role of the microcapsules used in limiting the intestinal absorption of the fluorescent MP microbeads. All the groups fed MP-implemented diets highlighted MP absorption at the intestinal level, as evidenced by the confocal microscopy analyses and, additionally, by a dense supranuclear vacuolization in the distal intestine (scattered in C, C/C+AX, and C+AX groups). Supranuclear vacuoles are known to be involved in the absorption through pinocytosis of intact nutrient molecules, especially in the distal tract of the fish intestine [115,116,117], and several studies demonstrated that microplastics can be internalized through this route of absorption [118,119]. The higher presence of supranuclear vacuoles in MP, MP/MP+AX, and MP+AX groups could be thus addressed as an indicator of effective MP internalization in the enterocytes. However, fish from the MP+AX group exhibited a significantly lower intestinal absorption of fluorescent MP microbeads compared to those from the MP group. This reduced absorption was reflected in a lower abundance of supranuclear vacuoles and in the reduced accumulation of dietary-derived microplastics in all the analyzed tissues/organs. Specifically, the absence of fluorescent MP microbeads in the muscular tissue of MP+AX fish was indeed a significant finding with implications for aquaculture safety. These outcomes can be interestingly explained by impaired microplastic absorption at the intestinal level due to the microencapsulation technology used for the AX administration. In fact, the in vitro coagulation test results indicated that both the microcapsules (whether empty or filled with AX) and starch, tested at the same concentration as in the experimental diets, effectively coagulated the fluorescent MP microbeads. This finding highlighted the potential impact of microencapsulation technology on the interaction between microplastics and biological systems, specifically in terms of absorption processes within the intestine. The absence of coagulation events in the solutions containing only Arabic gum or AX clearly evidenced that the starch present in the microcapsules has the main coagulating effect on the fluorescent MP microbeads. This result is in line with previous studies that used starch to coagulate water-suspended microplastics [94,120,121] and represents the first evidence of its same activity in the gut of a marine fish species. Particularly, starch has been reported to induce coagulation phenomena by bridging and enmeshing the suspended particles [94], with a mechanism that is particularly favored in acidic conditions [97,122,123], like those observed in carnivorous fish stomachs, in which the secreted HCl not only promotes digestion but also plays a functional role in improving starch performance as a coagulant [97]. In fact, HCl is regularly used to produce cationic starch, which shows, at pH 7 or below, a net positive charge [124], which is essential to increase the particle aggregation capacity of the starch compared to its native form [97]. In this regard, the in vitro coagulation events were observed both at pH 3 and 6, which are the average pH values of the stomach and intestine in European seabass [101], suggesting that these coagula were formed in the stomach and were then preserved during the intestinal transit. The discrepancy between the average size of the observed in vitro coagula (60 µm) and the reported absorption limit for MPs in the literature at the fish intestinal level (20 µm) thoroughly explained the reduced MP absorption in MP/MP+AX and MP+AX groups. The larger size of the coagula hindered the intestinal absorption of microplastics, leading to the observed outcomes in these experimental groups. This hypothesis was further supported by the number of fluorescent MP microbeads detected in the feces samples, which was significantly higher in both MP/MP+AX and MP+AX groups compared to the MP one.

## 5. Conclusions

The results obtained from the present study involving diets containing fluorescent MP microbeads and implemented with microencapsulated natural AX have shown promising outcomes for the aquaculture sector. These findings evidenced that microcapsules containing AX played a significant role in limiting the intestinal absorption of dietary MPs and were able to reduce oxidative stress response in fish. However, further studies are necessary to better understand the optimal timing and concentration of microencapsulated AX administration.

The authors suggest that the following steps should be conducted to depict a more comprehensive overview of the effectiveness of this technology in relation to the wide variability of conditions within the aquaculture sector. Particularly, the chemical interactions between microcapsules and microplastics should be further investigated by testing different types of polymers, water conditions (freshwater vs. salt water), dietary formulations, and gut environments in teleost (presence or absence of the stomach and, thus, of acidic digestion).

In each case, the present study introduced a valuable technology of great interest for the aquaculture sector due to its role in preventing (i) the microplastic absorption and translocation in farmed fish (with beneficial consequences also for the final consumer) by using natural molecules like starch which show a low cost, biodegradability, non-toxicity, and eco-friendliness, as well as (ii) the degradation of natural AX. Finally, even if the use of natural AX is presently affected by its high production costs, in light of the importance of lowering the carbon footprint, the exploitation of CO_2_-rich industrial effluents for the cultivation of algal biomasses promoting biogenic CO_2_ capture and utilization may represent an interesting solution for the industry and for the environment. This futuristic approach may thus promote a price reduction of both microalgae cultivation and AX production while promoting environmentally friendly activities.

## Figures and Tables

**Figure 1 antioxidants-13-00812-f001:**
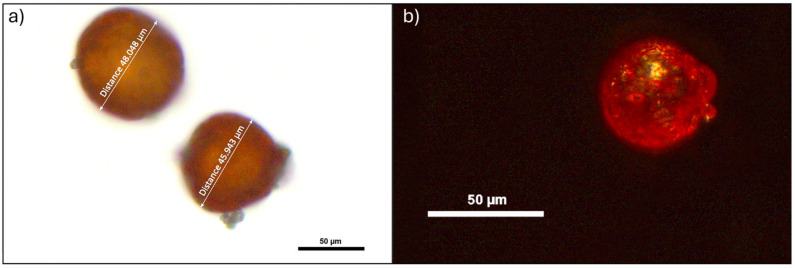
Details of AX microcapsules. (**a**) Cross section and (**b**) surface of the microcapsules used in the present study filled with AX. Scale bars: 50 µm.

**Figure 2 antioxidants-13-00812-f002:**
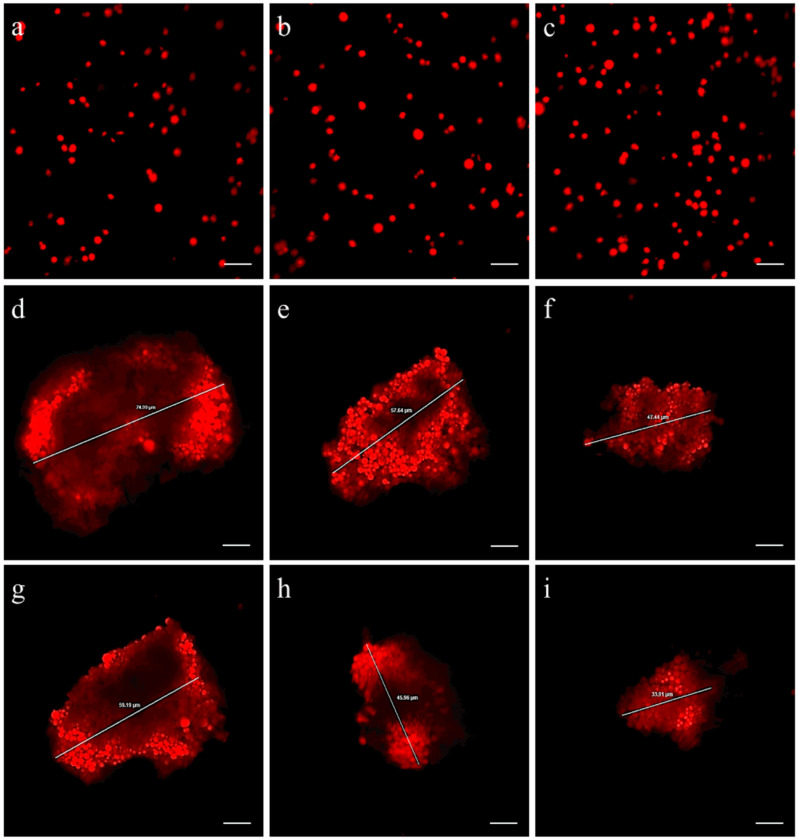
Examples of the different MP solutions tested in the in vitro coagulation experiment. First line: solutions containing (**a**) fluorescent MP microbeads alone or combined with (**b**) Arabic gum or (**c**) AX alone. Second line: example of coagula observed in solutions containing fluorescent MP microbeads combined with (**d**) AX microcapsules, (**e**) empty microcapsules, or (**f**) starch at pH 3. Third line: example of coagula observed in solutions containing fluorescent MP microbeads combined with (**g**) AX microcapsules, (**h**) empty microcapsules, or (**i**) starch at pH 6. Scale bars: 10 µm.

**Figure 3 antioxidants-13-00812-f003:**
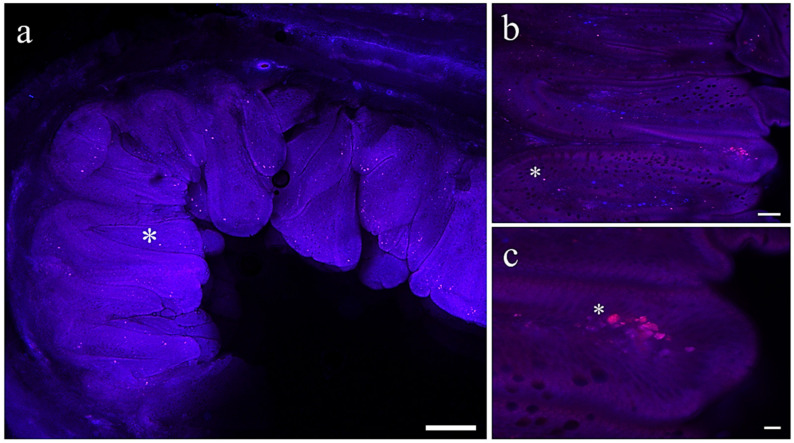
Representative images of intestine samples of European seabass from MP/MP-ASX group. Asterisks indicate an example of fluorescent MP microbeads. Scale bars: (**a**) 200 µm; (**b**) 50 µm; (**c**) 10 µm.

**Figure 4 antioxidants-13-00812-f004:**
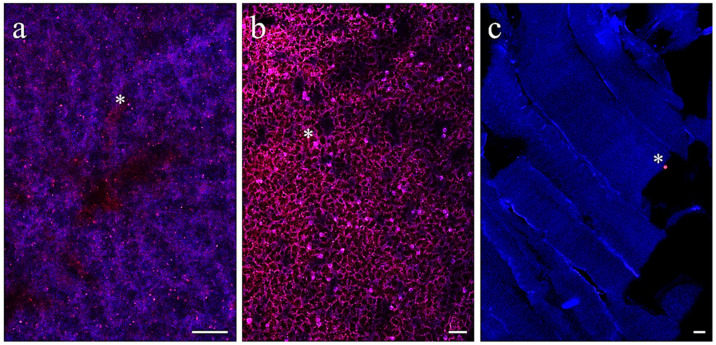
Representative images of (**a**) liver, (**b**) blood, and (**c**) muscle samples of European seabass fed the MP diet. Asterisks indicate an example of fluorescent MP microbeads. Scale bars: (**a**) 50 µm; (**b**) 20 µm; (**c**) 20 µm.

**Figure 5 antioxidants-13-00812-f005:**
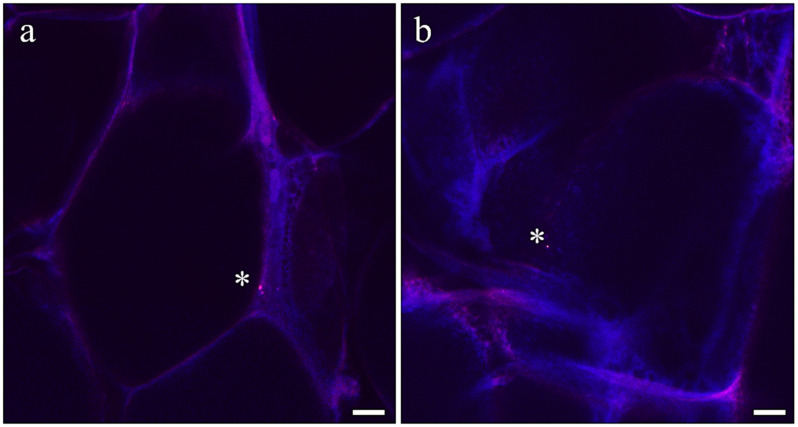
Representative images of (**a**,**b**) adipose tissue samples of European seabass fed the MP diet. Asterisks indicate an example of fluorescent MP microbeads. Scale bars: 10 µm.

**Figure 6 antioxidants-13-00812-f006:**
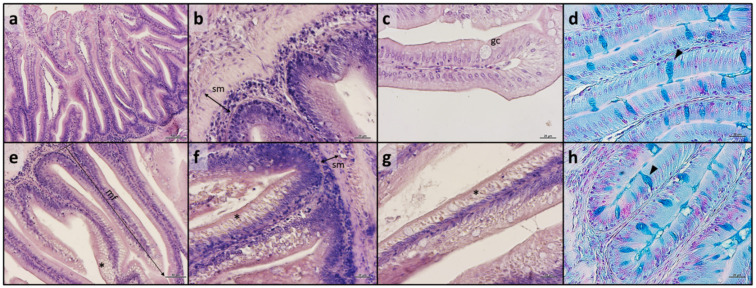
Example of histomorphology of distal intestine from European seabass fed the experimental diets. (**a**–**d**) different details of mucosal and submucosal architecture from fish fed C diet; (**e**) example of distal intestine sample from fish fed MP+AX diet; (**f**) focus on the basal portion of mucosal fold in fish from MP/MP+AX group; (**g**,**h**) details of mucosal architecture from fish fed MP diet. Abbreviations: sm, submucosa; gc, goblet cell; mf, mucosal fold. Symbols: arrowhead, Ab+ goblet cell; *, supranuclear vacuoles. Staining: (**a**–**c**,**e**–**g**) Mayer’s hematoxylin and eosin Y; (**d**,**h**) Alcian blue. Scale bars: (**a**) 100 µm; (**b**–**d**) 20 µm; (**e**) 50 µm; (**f**–**h**) 20 µm.

**Figure 7 antioxidants-13-00812-f007:**
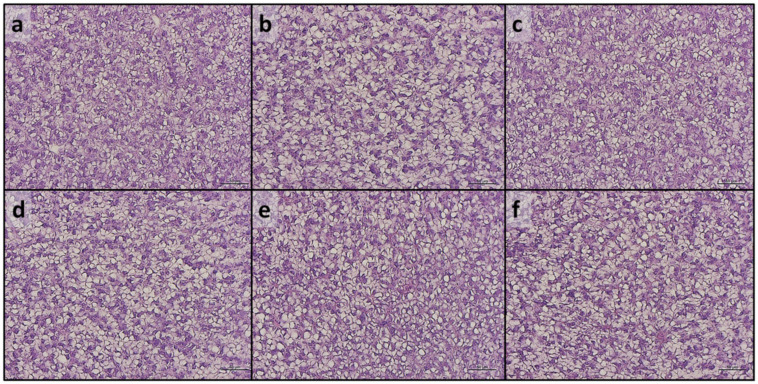
Example of histomorphology of liver from European seabass fed the experimental diets. (**a**) C; (**b**) C/C+AX; (**c**) C+AX; (**d**) MP; (**e**) MP/MP+AX; (**f**) MP+AX. Scale bars: 50 µm.

**Figure 8 antioxidants-13-00812-f008:**
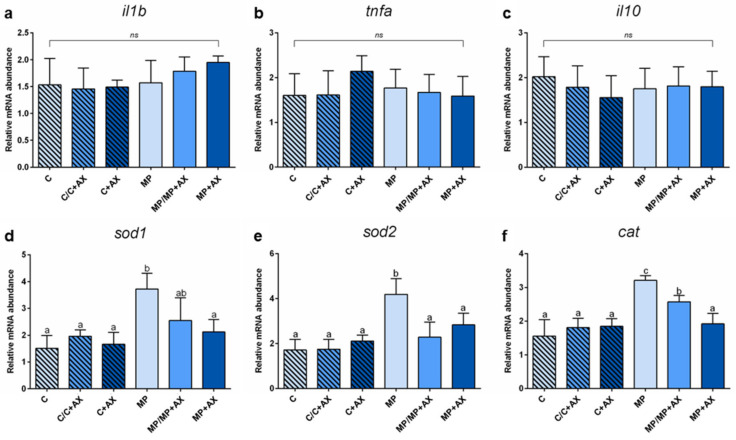
Relative mRNA abundance of genes involved in the immune response analyzed in distal intestine and in the oxidative stress response analyzed in liver of European seabass juveniles fed the different experimental diets. Values are presented as mean ± SD (n = 5). ^a–c^ different letters denote statistically significant differences among the experimental groups. ns, no significant differences (*p* > 0.05).

**Table 1 antioxidants-13-00812-t001:** Score assignment criteria for fusion episodes of mucosal folds, presence of supranuclear vacuoles in the enterocytes, Ab+ goblet cells’ relative abundance (on a 500 µm^2^ surface area), and lymphocyte infiltration at submucosal level in the distal intestine.

Index	Scores	Description
Episodes of mucosal folds (observations per section)	+	0–5
	++	5–15
	+++	>15
Supranuclear vacuoles	-	Absent
	+	Scattered
	++	Diffused
	+++	Highly abundant
Ab+ goblet cells	+	Scattered cells
	++	Diffused and widely spread
	+++	Highly abundant and tightly packed cells
Lymphocyte infiltration	+	Scarce
	++	Moderated
	+++	Diffused

**Table 2 antioxidants-13-00812-t002:** Forward and reverse sequences used in the present study, as well as annealing temperatures and NCBI IDs.

Genes	Forward Sequence (5′-3′)	Reverse Sequence (5′-3′)	AT (°C)	NCBI ID
*il1b*	AACTCCAACAGCGCAGTACA	AGACTGGCTTTGTCCACCAC	58	AJ_311925
*il10*	GCAGTCCCATGTGCAACAAC	TGCTACTGAACCTACGTCGC	59	AM_268529
*tnfa*	GACTGGCGAACAACCAGATT	GTCCGCTTCTGTAGCTGTCC	59	DQ_070246
*sod1*	AACCATGGTGATCCACGAGA	ATGCCGATGACTCCACAGG	60	FJ_860004.1
*sod2*	TGCCCTCCAGCCTGCTCT	CTTCTGGAAGGAGCCAAAGTC	58	MH_138007.1
*cat*	GGCTGGGAGCCAACTATCTG	GGAGCTCCACCTTGGTTGTC	58	MH_138006.1
*b-actin* (hk)	GGTACCCATCTCCTGCTCCAA	GACGTCGCACTTCATGATGCT	60	AJ_537421
*18s* (hk)	AGGGTGTTGGCAGACGTTAC	CTTCTGCCTGTTGAGGAACC	60	XM_051390998

Abbreviations: AT, annealing temperature; hk, housekeeping gene.

**Table 3 antioxidants-13-00812-t003:** Survival rate and growth indexes of European seabass juveniles fed the experimental diets.

	C	C/C+AX	C+AX	MP	MP/MP+AX	MP+AX	*p*-Value
IBW (g/fish)	74.6 ± 2.1 ^a^	74.7 ± 2.0 ^a^	74.9 ± 2.0 ^a^	74.5 ± 2.5 ^a^	74.8 ± 1.9 ^a^	74.1 ± 1.7 ^a^	0.955
FBW (g/fish)	146.8 ± 10.0 ^a^	144.5 ± 6.5 ^a^	143.3 ± 7.3 ^a^	146.4 ± 8.1 ^a^	144.0 ± 8.0 ^a^	146.3 ± 7.4 ^a^	0.863
WG (g/fish)	72.2 ± 8.0 ^a^	69.7 ± 4.7 ^a^	68.4 ± 4.3 ^a^	71.8 ± 5.7 ^a^	69.2 ± 6.2 ^a^	72.2 ± 5.3 ^a^	0.518
RGR (%)	96.5 ± 8.0 ^a^	93.2 ± 4.4 ^a^	91.2 ± 5.5 ^a^	96.2 ± 4.6 ^a^	92.4 ± 6.3 ^a^	97.3 ± 5.4 ^a^	0.081
SGR (%)	1.12 ± 0.07 ^a^	1.00 ± 0.04 ^a^	1.08 ± 0.05 ^a^	1.12 ± 0.04 ^a^	1.09 ± 0.05 ^a^	1.13 ± 0.04 ^a^	0.073

Values are shown as mean ± SD (n = 3). Abbreviations: IBW, initial weight; FBW, final weight; WG, weight gain; RGR, relative growth rate; SGR, specific growth rate. Superscript letters indicate statistically significant differences among the experimental groups.

**Table 4 antioxidants-13-00812-t004:** MP quantification in samples of intestine, blood, liver, muscle, and adipose tissue (microbeads/g) as well as in feces (microbeads/mg) of European seabass fed the experimental diets.

	C	C/C+AX	C+AX	MP	MP/MP+AX	MP+AX
Intestine	0	0	0	34.3 ± 4.7 ^a^	12.0 ± 3.0 ^b^	10.3 ± 1.5 ^b^
Blood	0	0	0	17.6 ± 2.8 ^a^	12.9 ± 1.5 ^b^	7.7 ± 1.5 ^c^
Liver	0	0	0	72.3 ± 5.8 ^a^	52.7 ± 11.4 ^a^	26.3 ± 8.0 ^b^
Muscle	0	0	0	3.0 ± 0.9 ^a^	3.2 ± 0.7 ^a^	0
Adipose tissue	0	0	0	7.1 ± 1.8 ^a^	3.7 ± 0.9 ^ab^	1.7 ± 0.6 ^b^
Feces	0	0	0	6461.5 ± 133.4 ^b^	7384.0 ± 127.6 ^a^	7476.9 ± 115.1 ^a^

Data are reported as mean ± SD (n = 15 for all the tissues; n = 10 for feces samples). ^a–c^ Within each line, different letters denote statistically significant differences among the experimental groups.

**Table 5 antioxidants-13-00812-t005:** Histological indexes measured in distal intestine of European seabass juveniles fed experimental diets.

	C	C/C+AX	C+AX	MP	MP/MP+AX	MP+AX
Mucosal fold height	566.1 ± 42.1	588.7 ± 35.5	569.1 ± 47.2	552.4 ± 30.4	583.6 ± 31.5	583.6 ± 44.3
Mucosal fold fusion	+	+	+	+	+	+
Supranuclear vacuoles	+	+	+	+++	+++	++
Ab+ goblet cells	++	++	++	++	++	++
Submucosa width	25.3 ± 2.1	25.9 ± 2.6	24.8 ± 2.3	26.7 ± 2.9	25.7 ± 2.6	26.3 ± 3.2
Lymphocytes infiltration	+	+	+	+	+	+

Values for mucosal fold height and submucosa width are shown as mean ± SD (n = 15).

## Data Availability

The original contributions presented in this study are included in the article. Further inquiries can be directed to the corresponding authors.

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
