# Peer review of "Mitigating Dietary Microplastic Accumulation and Oxidative Stress Response in European Seabass (Dicentrarchus labrax) Juveniles Using a Natural Microencapsulated Antioxidant"

_antioxidants, 2024, doi:10.3390/antiox13070812_

Round 1

Reviewer 1 Report

The manuscript includes a study focused on a very interesting item, a very important concern nowadays, i.e., the microplastic presence in the marine medium.

It includes a wide range of complementary analytical tools. I think it is well presented and justified and think it could be accepted for publication provided some minor aspects are clarified or performed.

Abstract

Include some concrete details of results obtained. As it is, this section provides a good overview of the work carried out but not about the real interest of the results.

Keywords

Include: “feeding trial” and “European sea bass”.

Introduction

Will the use of AX be economically possible ?

Material and methods

The number of replicates (n value) is not indicated. How many replicates were carried out for each experimental condition ?

Results and discussion

Table 3: Provide the statistical analysis.

Conclusions

Include some comment on the need for carrying out an optimisation design taking into account the feeding variables.

Some comment on the possible concern regarding the AX economical cost could be added.

Reviewer 2 Report

The paper deals with microplastic (MP) pollution in aquaculture studying the dietary MP accumulation in European seabass juveniles and proposes natural astaxanthin (AX) microcapsules to reduce MP absorption in fish. The topic is emerging and would attract the attention of the readers. The paper is organized in a rigorous manner and the data are clearly presented. In my opinion, there are three main limitations, that justify the revision of the manuscript before publication:

1) the choice of microplastic polymer

2) the lack of characterization of microcapsules

3) a few details of the mechanism of microplastic coagulation

1)     The introduction is lacking of a deep state of the art on the researches on mitigation of MP effects on fish.

2)     The authors used amino formaldehyde fluorescent microbeads as MPs for their experiments. This choice is unusual since the most frequently detected MPs in the aquatic environment are polyethylene (PE), polyethylene terephthalate (PET), polypropylene (PP), polyamide (PA), polystyrene (PS) and polyvinyl chloride (PVC). See for example doi.org/10.1007/s10924-022-02690-0 and other recent literature. The authors should justify this choice by also reviewing the literature data on the quantification of MP in fish.

3)     The description and characterization of microcapsules is completely missing but it is mandatory for understanding the potential applicability of the technique

4)     The size distribution of microbeads needs to be added, together with the same image of the surface and cross-section of the microcapsules

5)     The mechanism underlying  “microplastic coagulation” needs to be deeply analyzed and presented. Is this coagulation related to the specific MP types used in this research? Did the authors observe the same level of coagulation with other MPs, such as polystyrene or polyethylene?

6)     Why astaxanthin promotes better the MP coagulation?

7)     The authors should analyze with more detail the impact that their research could have on the issue of MP pollution in aquaculture.

Round 2

Reviewer 2 Report

The authors addressed the comments.

I have no other comments.